# Free and Microencapsulated Essential Oils Incubated In Vitro: Ruminal Stability and Fermentation Parameters

**DOI:** 10.3390/ani11010180

**Published:** 2021-01-14

**Authors:** Nida Amin, Franco Tagliapietra, Sheyla Arango, Nadia Guzzo, Lucia Bailoni

**Affiliations:** 1Department of Comparative Biomedicine and Food Science (BCA), University of Padua, Viale dell’Università, 16, 35020 Legnaro, Italy; nida.amin@uni-hohenheim.de (N.A.); sheylajohannashumyko.arangoquispe@phd.unipd.it (S.A.); nadia.guzzo@unipd.it (N.G.); 2Department of Agronomy Animals Food Natural Resources and Environment (DAFNAE), University of Padua, Viale dell’Università, 16, 35020 Legnaro, Italy; franco.tagliapietra@unipd.it

**Keywords:** essential oils, microencapsulation, rumen stability, in vitro incubation, fermentation parameters

## Abstract

**Simple Summary:**

Essential oils are an alternative for replacing antibiotics in animal feeds, but their volatile nature demands a high degree of stability. The aim of this study was to test the in vitro ruminal degradation of two different forms (free and microencapsulated) of three commercial products (mixtures of essential oils based on cinnamaldehyde, named Olistat-Cyn, Olistat-G, and Olistat-P) using an in vitro technique. The products were incubated in filter bags using an inoculum (buffer plus rumen fluid) for 48 h at 39 °C. It was found that the microencapsulation (matrix based on vegetable hydrogenated fatty acids) was efficient to protect essential oils from ruminal degradation because of the low disappearance of the microencapsulated essential oils in comparison to the free ones that were almost completely degraded. Olistat-G caused not only a significant decrease in the pH and the total protozoa number but also a significant increase in the total volatile fatty acids. As a conclusion, microencapsulation was found to be effective to ensure rumen by-pass and to be used as an additive in ruminant feeding. Among the essential oils tested, Olistat-G (mixture of cinnamaldehyde and vitamins) was capable of changing rumen fermentation, potentially reducing methane emissions.

**Abstract:**

Essential oils (EOs) are generally considered as an alternative to antibiotics because of their antimicrobial properties. Despite their vast variety, their volatile nature poses hindrance on their use in animal feeds, which demands a high degree of stability. This study aimed at testing the susceptibility of three EOs (mixtures of EOs based on cinnamaldehyde, named Olistat-Cyn, Olistat-G, and Olistat-P) in two forms (free: fEOs; and microencapsulated: mEOs) to in vitro ruminal degradation using the Ankom Daisy^II^ technique. The microencapsulation was made using a matrix based on vegetable hydrogenated fatty acids. Compared to the fEOs, which were completely degraded within 48 h of in vitro incubation, the mEOs showed a low ruminal disappearance. In comparison to the fermentation profile at 0 h, Olistat-G significantly decreased the pH and the total protozoa number after 48 h, while the total VFAs increased. However, the other EOs (Olistat-Cyn and Olistat-P) had no effect on the rumen fermentation parameters. In conclusion, the protection of EOs from ruminal degradation by microencapsulation was found to be very effective to ensure rumen by-pass. Among the EOs, Olistat-G was capable of changing rumen fermentation, potentially reducing methane emissions.

## 1. Introduction

The ban on antibiotics as growth promoters in animal feeds (1 January 2006) has resulted in an increase in demand for finding alternatives that could replace them. Over the last decade, a significant increase in studies on the use of natural plant extracts as antimicrobial growth promoters in animal feeds have been observed [1]. Essential oils (EOs) are volatile, aromatic plant extracts, with several antimicrobial properties such as degradation of bacterial cell walls, increase in membrane permeability, coagulation of cytoplasm, and destruction of cytoplasmic membranes and membrane proteins [2,3,4,5,6,7]. Owing to their wide range of antimicrobial properties, EOs are generally considered as potential alternatives to antibiotics [3]. Recently, an in vivo study [8] demonstrated that the use of a blend of EOs (CRINA^®^ Ruminants), alone or combined with enzymes (amylases), can be an alternative to ionophore antibiotics to improve ruminal fermentation and the milk yield and quality of dairy cows. Similarly, positive results were obtained [9] using EOs (a blend of thymol, eugenol, and vanillin) as a potential substitute for antibiotics to improve cattle performance during the finishing phase. Among the various EOs studied, cinnamaldehyde, thymol, and eugenol are the most commonly used in animal feeds because of their well-reported antimicrobial properties [10].

Cinnamaldehyde (3-phenylprop-2-enal) is the major component of cinnamon oil extracted from *Cinnamomum cassia* [11]. The antimicrobial activity of cinnamaldehyde is attributed to its protein-binding ability that provides protection from the action of amino decarboxylases [12,13,14]. Cinnamaldehyde inhibits the growth of certain bacteria, including *E. coli*, *C. perfringens*, *B. fragillis*, *B. breve*, *B. longum*, *L. acidophilus*, *L. reuteri*, and *S. serovars* [15,16].

Thymol (2-isopropyl-5-methylphenol) is found in thyme oil extracted from *Thymus vulgaris* [11]. Many studies have reported strong antiseptic and antibacterial properties of thymol due to its phenolic form, membrane disruption ability, and its role in the reduction of enzymatic activities [17,18,19,20,21]. Rota et al. [22] have shown growth inhibition of certain pathogenic bacteria, including *E. coli*, *E. coli* O157:H7, *L. monocytogenes*, *S. aureus*, *S. enteritidis*, *S. flexneri*, *S. sonnei*, *S. typhimurium*, and *Y. enterocolitica*, using thymol. In other studies, a dose-dependent effect of thymol on the activity of certain bacterial species, *B. animalis* ssp. *lactis*, *B. thermosphacta*, *C. perfringens*, *L. fermentum*, *S. enterica*, *S. serovars*, and *P. fluorescens*, have been discussed [23,24,25].

Eugenol (phenylpropene) is extracted from the essential oil of *Syzygium aromaticum* [10]. It possesses the ability of disrupting cytoplasmic membranes, which lead to a leakage of cellular contents, proteins, and ions, and ultimately could cause cell death [26]. The antibacterial spectrum of eugenol covers more than 24 bacterial genera, including the most commonly found pathogenic bacteria such as *E. coli* O157:H7, *E. coli* K88, and *Salmonella typhimurium* DT104, etc. [20,23].

In addition to the broad antibacterial properties of EOs, the antiparasitic activity of some EOs have also been highlighted. In an in vitro study by Remmal et al. [27], the lethal effects of thyme on oocysts of an intestinal parasite (*Eimeria*) have been reported. Similar antiparasitic activity against *Eimeria* infections and various other parasites of the gastrointestinal tract (*A. suum*, *T. suis*, and *O. dentatum*) has been observed using cinnamaldehyde [28,29]. Cinnamaldehyde can also modify the NF-κB and Nrf2 pathways to provide growth-promoting effects on the animals by acting as an antioxidant and anti-inflammatory agent [30]. Recently, antioxidant and anti-inflammatory responses were observed in lambs supplemented with a blend based on microencapsulated carvacrol, thymol, and cinnamaldehyde [31]. The immune-enhancing and antifungal properties of cinnamaldehyde and thymol also have been recently reported [9,32].

Despite the antibacterial, antifungal, antiparasitic, antioxidant, anti-inflammatory and immune-enhancing properties of a vast variety of EOs, their volatile nature poses hindrance to their use in animal feeds [9,31,33]. The majority of the cinnamaldehyde fed to the animals fails to reach the lower gut due to rumen microbial degradation or rapid absorption, particularly from the stomach and proximal duodenum. Furthermore, EOs may also be absorbed by the feed components [23]. As a result, the properties of the EO are affected [10]. Therefore, to remove the major obstacle for EO application in animal feeds, there is an urgent need to develop reliable methods to increase their stability, to ensure a site-specific release of the desired EOs in the gastrointestinal tract of the animals, and lower the effective dosage of the desired compounds to reduce the cost of the feed.

Microencapsulation is an emerging technology that is commonly used nowadays in human and animal nutrition for the preparation of stable products (vitamins, minerals, (polyunsaturated) fatty acids, and so on) [34]. Many studies have been conducted in the past on encapsulated EOs using different kinds of matrices to obtain the desired stability and site-specific, slow release of the products; e.g., microencapsulation with a lipid matrix is the most frequently used method for effective delivery of the product to the small intestine [35]. In a study by Zhang et al. [36], the use of alginate-whey protein microparticles for the site-specific delivery of EOs to the gut of broiler chickens have been reported.

Thus, the proper selection of the matrix material to obtain the desired degree of stability of the microencapsulated products is often a challenge and demands more research work. This study aimed at testing the susceptibility of three essential oils (EOs), namely, Olistat-Cyn, Olistat-G, and Olistat-P, in two different forms (free: fEOs; and microencapsulated: mEOs), to in vitro ruminal degradation using the Ankom Daisy^II^ technique. The ruminal stability of each product was assessed from the in vitro dry matter disappearance (DMD) values, calculated at six different incubation time periods. During this study, the effect of the fEOs on the modification of the fermentation parameters (pH, volatile fatty acids (VFAs), and protozoa number) was also studied.

## 2. Materials and Methods

### 2.1. Ethic Statement

All experimental procedures were carried out according to Italian law on animal care (Legislative Decree No. 26 of 14 March 2014) and approved by the ethical committee at the University of Padova (approval number 6/2020).

### 2.2. Essential Oils

This study was conducted using two different forms (free: fEOs; and microencapsulated: mEOs) of three essential oil (EO) products. The composition (per kg) of the microencapsulated products is reported below:Olistat-Cyn: 2b—cinnamaldehyde 20% (CAS no. 104-55-2); protected by a matrix of vegetable hydrogenated fatty acids, calcium carbonate, and wheat flour;Olistat-G: 2b—cinnamaldehyde (CAS no. 104-55-2), 3a710—vitamin K3 (1200 mg), and E330—citric acid 2000 mg; protected by a matrix of vegetable hydrogenated fatty acids, maize starch, calcium carbonate, and aromatic vegetable extracts;Olistat-P: 2b—cinnamaldehyde (CAS no. 104-55-2), 2b–thymol (CAS no. 89-83-8), and 2b–eugenol (CAS no. 97-53-0); protected by a matrix of vegetable hydrogenated fatty acids and calcium carbonate.

All EO products were provided by SILA s.r.l (Via Fermi Enrico n.1, Noale, VE, Italy). The products were stored at room temperature in dark aluminum bags and were subjected to the determination of loss from drying prior to their use in the tests.

### 2.3. Experimental Substrate

In order to provide a continuous energy source to the rumen microorganisms during in vitro incubation, a standard diet for dry cows was used as the substrate. The diet was added directly to each digestion vessel of an Ankom Daisy^II^ incubator (Ankom Technology, Macedon, NY, USA). The substrate consisted of corn silage, a mix of corn and barley meal, a mix of sunflower and soybean meal, alfalfa hay, ryegrass hay, sugar beet pulp, and other additives (291, 233, 173, 123, 120, 37, and 23 g/kg DM, respectively).

### 2.4. Rumen Fluid Collection and Processing

Two dry Italian Simmental cows were used as rumen fluid donors and were fed a specific diet consisting of hay ad libitum and 2.5 kg/d of concentrates (1 kg corn meal, 1 kg of sunflower meal, and 0.5 kg of dried sugar beet pulp). The diet was formulated to cover the nutritional requirements of dry cows and the animals were maintained on the same diet for at least 10 days prior to rumen fluid collection. Rumen fluid was collected before morning feeding using an esophageal probe [37]. The collected rumen fluid was poured into 2 thermal flasks preheated to 39 ± 0.5 °C, filtered through 4 layers of cheese cloth to eliminate feed particles, and immediately transferred to the department’s laboratory. In the lab the rumen fluid collected from the 2 cows was mixed and the pH was measured. All operations were conducted under anaerobic conditions by flushing with CO_2_, and the time required for all operations was less than 30 min.

### 2.5. Preparation of Buffer Solution

The buffer solution for in vitro incubation was prepared according to Holden [38], and it comprises Solution A (containing per 1 L distilled water: 10 g KH_2_PO_4_, 0.5 g MgSO_4_·7H_2_O, 0.5 g NaCl, 0.1 g CaCl_2_·2H_2_O, and 0.5 g urea), and Solution B (containing per 1 L distilled water: 15 g Na_2_CO_3_ and 1 g Na_2_S·9H_2_O). On the day of incubation, 266 mL of pre-warmed (39 °C) Solution B was added to 1330 mL of Solution A to obtain a ratio between Solutions A and B (1:5), followed by adjusting the pH to 6.8 with small additions (1 to 2 mL) of Solution B.

### 2.6. In Vitro Incubation

The in vitro incubation was conducted by allocating both the free and microencapsulated forms of each tested product to a single digestion vessel. The overall experiment consisted of three incubation runs per each tested product conducted over three consecutive weeks. During each incubation, three digestion vessels were prepared by adding 1596 mL of buffer solution (see above: “Preparation of buffer solution”), 400 mL of filtered rumen fluid (see above: “Rumen fluid collection and processing), and 15.96 g of experimental substrate, corresponding to 1 g substrate/100 mL buffer solution (see above:” Experimental substrate”). The EO products (0.2 g fEOs or 1.0 g mEOs) were separately placed in Ankom F57© filter bags (4.5 × 4.0 mm; 25 µm pore size; Ankom Technology, Macedon, NY, USA) in quadruplicates for each time-span. Filter bags were heat-sealed and added to the digestion vessels using fishing nets (to separate filter bags of free and microencapsulated products from each other with respect to different time periods). For each time point, 1 blank, heat-sealed filter bag (without product) was also added to each vessel to be used later for blank correction. Vessels were flushed with CO_2_ to create anaerobic conditions and incubated in a Daisy^II^ incubator (Ankom Technology, Macedon, NY, USA) at 39 °C for 0, 2, 6, 12, 24, and 48 h. A total of 540 filter bags were used in the experiment, as described below:-1 vessel for each EO product (Olistat-Cyn, Olistat-G, and Olistat-P) containing 60 filter bags: 2 forms (fEO + mEO) in quadruplicates for 6 incubation times (48 filled filter bags) and 2 blanks (filter bags without product) for 6 incubation times (12 empty filter bags);-3 vessels for each incubation run: 180 filter bags;-3 incubation runs: 540 filter bags.

For the fermentative parameters, the experimental design considered the 3 EOs (as the combined effect of the free and microencapsulated product incubated together in the vessel) × 2 incubation times (0 and 48 h of incubations) × 3 incubation runs.

### 2.7. Analytical Procedures

The removal of filter bags at the defined incubation times was very quick because the bags of each incubation time were inserted into a fishing net identifiable by the color of the ribbon that closed it. The filter bags were immediately washed with cold tap water until the water became clear, dried at 100 °C for 24 h, equilibrated for 15 min in a desiccator, and processed for the determination of in vitro dry matter disappearance (DMD) [39]. Fermentation fluid from each vessel was also analyzed at the start (0 h) and end of the incubations (48 h), to record the variations in pH, volatile fatty acids (VFAs), and protozoa number. For the determination of the VFAs, a 5 mL aliquot of fermentation fluids was collected at 0 and 48 h of incubation, added to 1 mL of (25%, w/v) metaphosphoric acid, and stored at −20 °C until analysis. For the analysis of the VFAs, frozen samples were thawed and centrifuged at 20 °C, 20,000× *g*, for 30 min. Supernatants were filtered through polypore 0.45 µm pore size filters (Alltech Italia, Milan, Italy) and the filtrate was injected into an HPLC (Thermo Finnigan SpectraSYSTEM, San Diego, CA), fitted with a SpectraSYSTEM P4000 pump, SpectraSYSTEM AS3000 auto sampler, and Waters 410 Differential Refractometer Detector. Separation was achieved using an Aminex HPX-87H column (300 mm × 7.8 mm; Bio-Rad) set at 75 °C. Data collection and integration was performed using ChromQuest software version 4.1 (Thermo Electron Corporation, San. Jose, CA, USA). The complete analysis of the VFAs in HPLC was performed by running an isocratic program for 30 min with 0.0025 N sulphuric acid as the mobile phase at a 0.6 mL/min flow rate. The identification of the analyte peaks was made by comparing the retention times of the samples with standard mixtures. The quantification of the VFA concentrations was dependent on the measurement of the peak area using an external standard method. The total number of protozoa in the fermentation fluids was determined by mixing 0.5 mL of the fermentation fluid with 0.5 mL of a saline formalin solution (NaCl 0.9%, formalin 4%), and dyeing the cells with a single drop of methylene blue, followed by quantification of the protozoa number with a Thoma cell counting chamber (0.1 mm depth, 1 × 1 mm square area of a large central square; Heinz Herenz Medizinalbedarf GmbH, Hamburg, Germany), using a light microscope with 20× magnification.

### 2.8. Statistical Analysis

Statistical analyses were conducted using the PROC GLM procedure in SAS (SAS Institute Inc., Cary, NC, USA, 2009) [40]. The data for dry matter disappearance (DMD, % DM) were analyzed with a model that considered the effect of the EO products (6 levels: three EOs in two different forms), the effect of the incubation time (six levels: 0, 2, 6, 12, 24, and 48 h), and the interaction EO products × incubation time. Data on the fermentation parameters (pH, protozoa number, and volatile fatty acids (VFAs)) were analyzed with a model that considered the effect of the EOs (three levels: combined effect of free and microencapsulated products of each EO incubated together in the vessel), the incubation times (2 levels: 0 and 48 h of incubation), and the interaction EO products × incubation time. The effect of each EO was tested by comparing the values measured at 0 and 48 h of incubation.

## 3. Results

### 3.1. Ruminal Stability of the EO Products

The ruminal stability of the EO products was assessed on the basis of their in vitro dry matter disappearance (DMD) values calculated at six different incubation times (Table 1).

All the free essential oils (fEOs) showed a similar pattern: an increase in the DMD values with incubation time, with maximum disappearance of the products observed after 48 h of incubation: 64.0, 80.5, and 84.6% for Olistat-Cyn, Olistat-G, and Olistat-P, respectively.

In contrast to the fEOs, the microencapsulated essential oils (mEOs) were highly stable in the ruminal environment. The water-soluble part of the matrix material of the mEOs was already dissolved in the water after 0 h of incubation, giving similar solubility values (12.0, 14.4, and 17.7 for Olistat-G, Olistat-P, and Olistat-Cyn, respectively). The mEO products showed relatively low or no disappearance in the ruminal environment, as indicated by the low DMD values after 48 h: 12.4, 13.4, and 13.7% for Olistat-P, Olistat-Cyn, and Olistat-G, respectively.

In addition, the differences between the DMD values of the free and microencapsulated form of the same product during different time intervals were highly significant (*p* < 0.05), excluding the comparison at 0 h for Olistat-Cyn (20.9 vs. 17.7%; *p* = 0.989) and Olistat-P (18.3 vs. 14.4%; *p* = 0.809).

### 3.2. Rumen Fermentation Parameters and Protozoa Number of EO Products

The fermentation fluid of each digestion vessel was analyzed at the start (0 h) and the end (48 h) of incubation to observe possible modifications in the rumen fermentation parameters and protozoa number (Table 2). In each vessel, the free and microencapsulated products of each essential oil were incubated together. However, due to the high ruminal stability of all mEOs, it is possible to assume that most of the effects on the rumen fermentation parameters and protozoa number depend on the essential oils released from the solubilization of the fEO products.

In comparison to the fermentation profile at 0 h, Olistat-G caused a significant decrease in the pH (6.93 vs. 6.36; *p* < 0.001) and total protozoa number (4.73 vs. 4.40 log10/mL; *p* < 0.05) after 48 h, while a significant increase in the total VFAs (17 vs. 33 mmol/L; *p* < 0.05), with an increase in the proportion of n-valerate (0.63 vs. 1.30%; *p* < 0.05) and a subsequent decrease in the proportion of iso-valerate (1.30 vs. 0.47%; *p* < 0.001), was observed. However, no significant differences (*p* > 0.05) were detected between the molar proportions of other individual VFAs (acetate, propionate, butyrate, iso-butyrate, and caproate) at 0 and 48 h.

Olistat-P after 48 h of incubation also caused a significant decrease in the pH (6.93 vs. 6.46; *p* < 0.001), but no modification in the protozoa number and fermentation parameters, excluding a decrease in the proportion of iso-valerate (1.30 vs. 0.54%; *p* < 0.001).

No statistical differences were observed for Olistat-Cyn after 48 h of incubation in the pH, total protozoa number, total VFAs, and proportion of the individual VFAs, excluding, as reported for all EO products, a decrease of the percentage of iso-valerate (1.30 vs. 0.73%; *p* < 0.001).

## 4. Discussion

The results obtained in this study indicated that encapsulation of essential oils with a matrix based on vegetable hydrogenated fatty acids and calcium carbonate, with or without the addition of wheat flour or maize starch or aromatic vegetable extracts, was efficient in protecting the essential oils from ruminal degradation. This result is confirmed by the minimal disappearance of the microencapsulated products from the filter bags in comparison to the free products that were almost completely degraded within 48 h of in vitro incubation in the ruminal environment. To the best of our knowledge, this study is the first of its type to show the comparison between the ruminal stability of free and microencapsulated forms of different essential oil products containing either a single essential oil or a blend of different essential oils (cinnamaldehyde, thymol, and eugenol), vitamins, or pro-vitamins. Kim et al. [34], in a recent in vitro study, reported a high protection effect (99%) from rumen microbes for linseed oil coated with hydrogenated palm oil after 8 h of incubation.

Essential oils are generally considered as potential alternatives to antibiotics owing to their wide range of antimicrobial, anti-inflammatory, anti-parasitic, and immune-enhancing properties. However, the use of EOs as antimicrobial growth-promoting agents in animal feedstuff demands a high degree of stability from them. Without proper protection, there is a high probability that the majority of their components will be lost during the manufacture and processing of the feedstuff, or degraded, digested, and inactivated in a gut tract (rumen, stomach, or intestine) different from the desired action site [41]. This reduces the amount and the kinetic of action of the EO reaching the lower digestive tract of the animal and, as a result, the beneficial effect of the EOs as well as feed profitability will be affected [10,42].

Many studies have reported the influence of feed composition on the activity of EOs. Feedstuffs rich in proteins and fats may protect the bacterium from EO activity [43,44,45]. Si et al. [41] observed a reduction in the antimicrobial activity of EOs due to absorption and binding to the feed components, which made the essential oils unavailable to the target pathogens. In the case of monogastric feeding, Michiels et al. [7] observed a decrease in the antimicrobial activity of cinnamaldehyde and eugenol due to rapid absorption and degradation of the major part of the product in the stomach. In addition, the majority of EOs are chemically unstable and oxidize rapidly when they come in contact with oxygen, moisture, heat, and light [46,47,48]. This oxidation of EOs results in the production of free radicals and a reduction in their antimicrobial activity, shelf-stability, quality, and nutritional value [49].

As a result of the volatile nature and low stability of EO products, the demand for manufacturing stabilized EO components is increasing. Microencapsulation of EOs is an efficient approach to ensure the functional and biological characteristics of EOs, protecting them from environmental oxidation reactions and increasing their stability to allow controlled release of the product [48,50,51,52]. Many studies have reported better antimicrobial activity of EOs by protecting them from degradation using microencapsulation technology. In a study by Si et al. [41], encapsulation using fenugreek gum was found to be efficient in maintaining the antimicrobial activity of cinnamaldehyde during storage. Ayala-Zavala et al. [53] also observed better antifungal activity of cinnamon leaf oil against *Alternaria alternata* using B-cyclodextrin as the encapsulation material.

In addition to highlighting the ruminal stability of the EO products and the benefits of using microencapsulation technology for protecting essential oils from ruminal degradation, this study has also reported the effect of the EO products on rumen fermentation parameters and protozoa number. The results of this study show that, in comparison to the fermentation profile at 0 h, only Olistat-G (a commercial blend of cinnamaldehyde, vitamin K3, and citric acid) caused a significant decrease in the pH and the total protozoa number, with a significant increase in the total VFAs after 48 h. On the contrary, the other two EO products, Olistat-Cyn (cinnamaldehyde 20%) and Olistat-P (a commercial blend of cinnamaldehyde, eugenol, and thymol), caused no significant modifications in the rumen fermentation parameters and protozoa number within 48 h of incubation. This mixture of different compounds in Olistat G is probably able to selectively stimulate ruminal bacteria and improve their efficiency in using the different substrates to obtain VFAs as end-products of the fermentation and, at the same time, to have a negative effect on the population of protozoa. To our knowledge, there are currently no published in vivo or in vitro experiments available on the effect of different combinations of essential oils with vitamins and preservatives to understand the mechanisms of action of these compounds at the rumen level.

Olistat-Cyn is a commercial product containing 20% cinnamaldehyde. The absence of effects of this product on the rumen fermentation parameters is in accordance with the results of other studies that used cinnamaldehyde. In an in vitro study by Benchaar et al. [54], using 400 mg/L of cinnamaldehyde, no effect was observed on rumen pH, total and individual volatile fatty acids (VFAs), ammonia, and gas production. In a short-time study [33], the rumen microbial protein synthesis would be reduced when the dairy cows are supplemented with cinnamaldehyde at a dosage of 2 mg/kg of BW. The results of the in vitro study were confirmed further in an in vivo assay by supplementing the lactating cows’ diets with cinnamaldehyde (1 g/day). The cinnamaldehyde supplementation had no effect on the rumen fermentation parameters, protozoa number, nor dry matter intake [55]. In another in vivo study, where lambs’ diets were supplemented with 200 mg/kg DM of cinnamaldehyde, no effects were observed on the dry matter intake, total VFAs, ammonia, or rumen pH [56]. Similarly, Macheboeuf et al. [57] showed a negative impact of higher cinnamaldehyde doses on rumen fermentation parameters. In a recent study, cinnamaldehyde was reported to be a non-viable candidate for methane mitigation strategies in dairy cows as it had no effect on in sacco ruminal degradation, dry matter intake, modification of rumen fermentation parameters, protozoa number, or rumen pH [58].

In the present study, Olistat-P, a commercial blend of cinnamaldehyde, thymol, and eugenol, caused a significant decrease in pH (*p* < 0.001) after 48 h of incubation. However, no significant effects of this product were observed on rumen protozoa number, nor for individual and total VFAs. Few studies have been conducted in the past to show the combined effect of different blends of EOs on rumen fermentation parameters and this area needs to be explored further. Cardozo et al. [59] performed an in vivo assay by supplementing heifers’ diets with a mixture of 600 mg/day of cinnamaldehyde and 300 mg/day eugenol, and observed a decrease in the acetate proportion and ammonia concentration, but an increase in the propionate proportion. However, the total VFAs, rumen pH, and butyrate proportion remained unaffected by the treatment. Tager and Krause [60] showed that supplementation of dairy cows’ diets with a mixture of cinnamaldehyde (85 mg–1.7 g/day) and eugenol (140 mg–2.8 g/day) had no effect on the dry matter intake, rumen pH, ammonia, VFA concentrations, and acetate/propionate ratio. In a recent study by Al-Saht et al. [61], using a blend of thymol and cinnamaldehyde, no effects on the rumen parameters and protozoa number were observed. Recently, De Souza et al. [9] reported that the in situ digestibility values of DM, crude protein, and NDF were similar in heifers fed diets without or with a protected blend (4 g/animal/d) of essential oils (eugenol, thymol, and vanillin). The results of these studies are highly variable and dependent on various factors, such as the type of EO used in the blend, the rumen microbial community, pH, and the length of adaptation period to these EOs by the rumen bacteria [62].

Furthermore, using a blend of EOs is often advantageous compared to using a single EO. Friedman et al. [63] showed that cinnamaldehyde can be protected from heat-induced destruction by mixing it with eugenol. In order to make an EO product effective so that it can potentially replace antibiotics, the use of an EO blend is often preferred, since it is unlikely that any individual EO can cover all possible properties of antibiotics. Moreover, there may be some combined effect between different EOs, which could possibly reduce the required effective dosages. Some studies have already reported the beneficial effects of the use of different blends of antibiotic alternatives on the growth performance and health of weaned pigs [64,65]. In addition, the use of encapsulated EO blends is preferable to that of a liquid EO blend because the covering matrix may provide an intra-ruminal slow release of the encapsulated blend of EO and can minimize the adverse effects caused by a high dosage [66].

From a practical point of view, Olistat-G, the blend of cinnamaldehyde, vitamin K3, and citric acid, was capable of changing in vitro rumen fermentation, increasing the VFA production and reducing the protozoa number, potentially mitigating the methanogenic Archaea, which are either ectosymbionts or endosymbionts in protozoa cells. These results will have to be confirmed in an in vivo experiment to verify also the effects on the performance and on the health of the animals.

## 5. Conclusions

In conclusion, the protection of EOs from ruminal degradation by microencapsulation, based on hydrogenated fatty acids, calcium carbonate, and starch, was found to be very effective in reducing rumen microbial degradation. All the tested products showed a high stability in the ruminal inoculum, with similar values from 86.3 to 87.6% of DM after 48 h of in vitro incubation. Therefore, the EOs contained in the microspheres can by-pass the rumen and reach the gastro-intestinal tract and carry out their beneficial action.

In addition, the pattern of the free products during fermentation suggests one should consider the role of essential oils on the rumen’s microbial population. Among the tested products, Olistat-G was capable of changing rumen fermentation, reducing the protozoa number, and increasing VFA production, with potential beneficial effects such as reducing methane emissions. However, in vivo experiments are needed to verify the dosage and long-term effects of this EO in ruminants.

## Figures and Tables

**Table 1 animals-11-00180-t001:** LS means of the in vitro dry matter disappearance (DMD, % DM) of the essential oil products, free (fEO) or microencapsulated (mEO), after incubation.

Product	Olistat-Cyn				Olistat-G				Olistat-P			
Form ^1^	fEO	mEO	SE ^2^	*p*-Value	fEO	mEO	SE ^2^	*p*-Value	fEO	mEO	SE ^2^	*p*-Value
Incubation time (h)												
0	20.9	17.7	1.9	0.989	56.0	12.0	2.1	<0.001	18.3	14.4	1.5	0.809
2	26.4	17.5	1.8	0.034	75.6	12.7	1.7	<0.001	36.3	12.4	1.8	<0.001
6	30.0	17.7	2.4	0.023	76.1	13.0	2.1	<0.001	53.8	14.4	2.2	<0.001
12	50.4	17.9	2.4	<0.001	76.6	13.0	1.9	<0.001	55.3	12.4	2.8	<0.001
24	50.7	14.1	1.9	<0.001	80.2	13.6	1.7	<0.001	55.9	12.3	1.7	<0.001
48	64.0	13.4	1.4	<0.001	80.5	13.7	1.2	<0.001	84.6	12.4	1.8	<0.001

^1^ fEO = free essential oil; mEO = microencapsulated essential oil. ^2^ SE = standard error of the LS mean.

**Table 2 animals-11-00180-t002:** LS means of the pH, protozoa number, and volatile fatty acids (VFA) profile, observed at 0 and 48 h of incubation of the essential oil products.

Product	None	Olistat-Cyn	Olistat-G	Olistat-P	RMSE ^1^	*p*-Value
Incubation Time (h)	0	48	48	48		
pH	6.93 ^B^	6.73 ^AB^	6.36 ^A^	6.46 ^A^	0.12	<0.001
Protozoa (log10/mL)	4.73 ^b^	4.72 ^ab^	4.40 ^a^	4.56 ^ab^	0.19	0.099
Total VFA (mmol/L)	17 ^a^	18 ^a^	33 ^b^	27 ^ab^	9.8	0.095
VFA (%)						
Acetate	71.7	70.4	71.6	72.4	4.5	0.96
Propionate	16.6	18.2	18	16.1	3.5	0.82
n-butyrate	8.4	9.1	7.9	9.7	1.8	0.61
Iso-butyrate	0.79	1.00	0.34	0.67	0.53	0.51
n-valerate	0.63 ^a^	0.39 ^a^	1.30 ^b^	0.47 ^a^	0.31	0.012
Iso-valerate	1.30 ^B^	0.73 ^A^	0.47 ^A^	0.54 ^A^	0.25	<0.001
n-caproate	0.47	0.25	0.40	0.15	0.24	0.24
Acetate/Propionate ratio	4.31	3.88	4.67	4.69	1.0	0.71

^1^ RMSE = root mean square error; ^a,b^: different superscripts within a row indicate the means differ (*p*
≤ 0.05); ^A,B^: different superscripts within a row indicate the means differ (*p*
≤ 0.001).

## Data Availability

The data presented in this study are available on request from the corresponding author.

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
