# Peer review of "Free and Microencapsulated Essential Oils Incubated In Vitro: Ruminal Stability and Fermentation Parameters"

_animals, 2021, doi:10.3390/ani11010180_

Round 1
Reviewer 1 Report
I advise to define the composition of microencapsulated products, including the ratio of active ingredient to coating material.
In addition, a new group with coating substances addition should be considered as corrections for microencapsulated product.
Reviewer 2 Report
The present study investigates the degradation of essential oils in rumen either in their natural form or encapsulated. In vitro trials usually do not accurately reflect the effect of such feed additives compared to in vivo ones and lack of representativeness. However, the present study has been conducted correctly, the methods are well-presented and the results have been discussed in depth. In addition, the Daisy technique is an “old-fashioned” procedure, but the literature lacks of such information and in my opinion, this is a great aspect of the submitted paper which provides further knowledge for the aforementioned technique.
General comments:
- Since the manuscript and foremost the M&M section is not extensive, I would like to see more information about the In vitro trial. More specifically, in line 142, authors could provide further information about buffer and rumen fluid collection additional to references. The way that in vitro digestibility was conducted is the core of your work. Please highlight it.
- In table 2 and specifically in total VFA I would suggest you to crosscheck the significant superscripts. More specifically, 17 vs 33 is significant but 18 vs 33 is not. I am aware of the high SD but even if this is just a tendency (p=0.095), I would like to see an explanation about the rumen biochemistry underling these alternations between Olistat-Cyn, Olistat-G, and Olistat-P.
- In discussion section, I would suggest you to provide a short assumption based on your results of how these feed additives would affect the aforementioned investigated parameters in an in vivo trial and what have been done up to now.
- Last but not least, the cited literature is quite old, specifically, only 18 out of 59 citation have been published the last 10 years and only 2 citation have been published the last 5 (2 publication of 2016). The average cited references of the discussion section are from 2006. In my opinion, you should improve the introduction and discussion with newest research findings.
Reviewer 3 Report
The submitted paper presents a well organized and performed in vitro experiment.
The idea of use encapsulated essential oils as special additives in ruminant’s nutrition is of big interest, regarding the special conditions in rumen.
The laboratory work presented here is complete, with certain design and appropriate use of methods (laboratory and statistic) to complete it. Results clearly presented and conclusions are justified from the content of the paper. English text is very carefully edited without mistakes of incomprehensive parts. A correction is needed in L306 (a verb is missing).
Overall the paper offers a significant proof of the essential oils use in ruminants, and a step for further studies, which should be in vivo in real conditions.
As a drawback for the paper it could be mentioned that it is of limited reader’s interest, mainly due to the in vitro and “artificial” situation it deals.
Reviewer 4 Report
The paper is interesting, material and methods adequately described.
However it is not clear how many filter bags you used for each vessel in each incubation. You said that you used three digestion vessels, allocating both the free and microencapsulated forms of each tested product (three products) in quadruplicate for 6 different times. Daisy Incubator could contain 24 filter bags/vessel. Can you clarify.
Moreover when you removed filter bags from digestion vessel at different incubation times, you open the vessel and then you flushed them again with CO2? Could you described.
In discussion the product Olistat G is few discussed. Could you compare your results with references.
Line 268 Si et al [34]
Line 286 (1 g/day)
Line 326 gastro-intestinal
In references, the pages should be written in italics. ".......J. Food Sci. 1984, 49(2),...."
Round 2
Reviewer 2 Report
Dear Authors,
The manuscript was improved. Please see the following comments on your replies.
1)
R: Since the manuscript and foremost the M&M section is not extensive, I would like to see more information about the In vitro trial. More specifically, in line 142, authors could provide further information about buffer and rumen fluid collection additional to references. The way that in vitro digestibility was conducted is the core of your work. Please highlight it.
AU: Thank you. Comment accepted. The text has been modified
About the buffer solution: a specific paragraph has been included in the revised manuscript (lines 151-157).
About the rumen fluid collection: the rumen liquor was collected following the procedures detailed
by Tagliapietra et al. [37]. More details on how rumen fluid was collected, stored and transferred to
the laboratory have been added (lines 145-150).
R: Thank you to follow this suggestion. The article was improved.
2)
R: In table 2 and specifically in total VFA I would suggest you to crosscheck the significant superscripts. More specifically, 17 vs 33 is significant but 18 vs 33 is not. I am aware of the high SD but even if this is just a tendency (p=0.095), I would like to see an explanation about the rumen biochemistry underling these alternations between Olistat-Cyn, Olistat-G, and Olistat-P.
AU: Yes, thank you. We checked the differences within the row of total VFA and we verified the superscripts of Olistat-Cyn was incorrect (see Table 2 of the revised manuscript).
An attempt to explain the fermentation results has been included in the discussion section (lines 309-315 of the revised manuscript).
R: This point was improved. Please, update the results section in line 247 with your updated data. Specifically, state that Olistat-G found out to produce higher amounts of VFA than that of Control and Olistat-Cyn.
3)
R:In discussion section, I would suggest you to provide a short assumption based on your results of how these feed additives would affect the aforementioned investigated parameters in an in vivo trial and what have been done up to now.
AU: Suggestion accepted. At the end of the discussion section a final paragraph has been included in order to suggest Olistat-G as the best additive on the basis of the in vitro fermentation parameters. The need to verify these results in vivo has been also reported (lines 360-361 of the revised manuscript).
R: Thank you to follow the suggestion. However, protozoal decline is not only accompanied by a reduction in methanogens population, but also with a significant decrease in OM, NDF, ADF digestibility. For this reason, it is not a good idea to "praise" protozoa defaunation, since may conceals a lower feed efficiency on farm scale. Thus, I suggest you to "trim" your statement as:
"From the practical point of view, Olistat-G, the blend of cinnamaldehyde, vitamin K3 and Citric acid, was capable of changing in vitro rumen fermentation increasing the VFAs production and reducing protozoa number with potential effects on mitigation of methanogenic Archaea which are either ectosymbionts or endosymbionts in protozoa cells.
Newbold, C.J.; de la Fuente, G.; Belanche, A.; Ramos-Morales, E.; McEwan, N.R. The Role of Ciliate Protozoa in the Rumen. Microbiol. 2015, 6. doi:10.3389/fmicb.2015.01313.
4)
R: Last but not least, the cited literature is quite old, specifically, only 18 out of 59 citation have been published the last 10 years and only 2 citation have been published the last 5 (2 publication of 2016). The average cited references of the discussion section are from 2006. In my opinion, you should improve the introduction and discussion with newest research findings.
AU:We agree with an update of the literature, as recommended by the Reviewer. Seven new citations have been included in the references (1, 2, 8, 9, 31, 33, 34). Four papers are published in 2020, two in 2019, one in 2018.
R: Thank you to follow this suggestion. The article was improved.
Please use subscripts in chemicals (line 153).
